# Evidence of potential overdiagnosis and overtreatment of attention deficit hyperactivity disorder (ADHD) in children and adolescents: protocol for a scoping review

Luise Kazda [1], Katy Bell [1], Rae Thomas [2], Kevin McGeechan,[1] Alexandra Barratt[1]

¹Sydney School of Public Health, The University of Sydney, Sydney, New South Wales, Australia
²Institute of Evidence-Based Healthcare, Bond University, Gold Coast, Queensland, Australia

**Correspondence to**
Luise Kazda;
luise.kazda@sydney.edu.au

## ABSTRACT

**Introduction** Worldwide, attention deficit hyperactivity disorder (ADHD) diagnosis rates in children and adolescents have been increasing consistently over the past decades, fuelling a debate about the underlying reasons for this trend. While many hypothesise that a substantial number of these additional cases are overdiagnosed, to date there has been no comprehensive evaluation of evidence for or against this hypothesis. Thus, with this scoping review we aim to synthesise published evidence on the topic in order to investigate whether existing literature is consistent with the occurrence of overdiagnosis and/or overtreatment of ADHD in children and adolescents.

**Methods and analysis** The proposed scoping review will be conducted in the context of a framework of five questions, developed specifically to identify areas in medicine with the potential for overdiagnosis and overtreatment. The review will adhere to the Joanna Briggs Methodology for Scoping Reviews. We will search Medline, Embase, PsycINFO and the Cochrane Library electronic databases for primary studies published in English from 1979 onwards. We will also conduct forward and backward citation searches of included articles. Data from studies that meet our predefined exclusion and inclusion criteria will be charted into a standardised extraction template with results mapped to our predetermined five-question framework in the form of a table and summarised in narrative form.

**Ethics and dissemination** The proposed study is a scoping review of the existing literature and as such does not require ethics approval. We intend to disseminate the results from the scoping review through publication in a peer-reviewed journal and through conference presentations. Further, we will use the findings from our scoping review to inform future research to fill key evidence gaps identified by this review.

## INTRODUCTION
### Review questions
Our primary study question for this scoping review is: 'Does the published literature indicate a potential for overdiagnosis and/or overtreatment of Attention Deficit Hyperactivity Disorder (ADHD) in children and adolescents?' Besides this question, we will consider two further, secondary questions in our review: 'When mapped against an existing framework for identifying potential overdiagnosis,[1] are there any gaps in the current evidence?' and 'How does the framework[1] perform in terms of applicability and usability for identifying potential overdiagnosis of a mental health condition?'

### Background
With steadily increasing prevalence rates of ADHD throughout the developed world,[2–6] there is growing debate about whether this trend is due to an actual increase in prevalence, better detection and diagnosis, misdiagnosis, or overdiagnosis.[7–10] While the evidence for overdiagnosis in many other conditions (especially in screen-detected cancers) is increasingly recognised,[1 11 12] the evidence for overdiagnosis of ADHD, a non-cancer condition where overdiagnosis

### Strengths and limitations of this study

► This is the first comprehensive synthesis of evidence on the potential for overdiagnosis and overtreatment of attention deficit hyperactivity disorder in children and adolescents.
► Broad and systematic search strategy is used to retrieve all relevant studies published since 1979.
► The review compares evidence against established criteria for potential overdiagnosis and overtreatment.
► The review includes a critical appraisal of included studies but no formal risk of bias assessment.
► Underdiagnosis and undertreatment may also occur, but are outside the scope of this review.

is widely thought to occur, has not yet been comprehensively evaluated.[13] Quantifying the potential for overdiagnosis is only just emerging as a field of interest in both paediatric and mental health research.[7 14]

The potential overdiagnosis of ADHD could be caused by three main drivers. First, it can arise due to the problem of overdefinition,[15] that is, lowering the threshold for a disease, by expanding the disease definition to include people with ambiguous or very mild symptoms without evidence that doing so improves patients' health overall and in the longer term.[16 17] Second, overdiagnosis may also be caused by overdetection[4 18] (eg, screening children at young ages for behaviour problems), and third by the medicalisation of some behaviour patterns (eg, those typical of relatively younger school children).[19] Other factors may also have played a part. These include formal changes to the diagnostic threshold (Diagnostic and Statistical Manual of Mental Disorders (DSM)-5 vs DSM-IV), pharmaceutical industry influence,[13] and health and social service drivers (eg, access to resources linked to a diagnosis).[10]

Although the prevalence rates of ADHD seem to have risen substantially in the past few years, most cases continue to be reported as mild to moderate forms (87% in 2007, 84% in 2011 and 86% in 2016 of all reported cases in a large USA-based survey),[2 20] and it has been argued that many of those cases may represent overdiagnosis. Should this be the case, these children may not experience a net benefit from an ADHD diagnosis and subsequent treatments, but may be harmed.[13]

While correctly diagnosing and treating ADHD have many potential benefits,[21] the harms from overdiagnosing and overtreating ADHD are significant and costly on multiple levels. Not only may the individual child experience negative physical and psychosocial effects, their families may also experience psychosocial and financial burdens. Overdiagnosis and overtreatment also result in financial and opportunity costs to the health system, and to society at large.[7] It has also been pointed out that the resulting potential overuse of healthcare resources contributes to the simultaneous underuse of said resources,[22] for example, by depriving underdiagnosed and undertreated groups of children who would largely benefit from ADHD treatment and timely access to diagnostic and treatment services.[23] While also addressing the twin issues of underdiagnosis and undertreatment is beyond the scope of this proposed review, we see our work as a starting point for a broader discussion around the principles of 'right care', where resources need to be reallocated to where they are most needed and effective.[22]

### Rationale

In summary, while there are increasing concerns about the potential for overdiagnosis of ADHD in children and adolescents, there is scant evidence quantifying the problem. Consequently, a systematic review of the literature to quantify ADHD overdiagnosis is not possible due to the current lack of synthesisable evidence in this field. The rationale for this broader scoping review then is to use a recently developed framework of questions[1] to systematically determine if the existing literature indicates a potential for overdiagnosis and overtreatment in ADHD. We hypothesise that ADHD fulfils these predetermined criteria for potential overdiagnosis. A secondary aim is to further examine and highlight any gaps in the current evidence that may prevent us from determining whether or not ADHD is overdiagnosed and overtreated. This aim will be especially helpful in guiding subsequent research in this area. A further secondary aim of the study is to rigorously test the existing five-question framework[1] for its applicability and usability in another area, namely paediatric mental health.

Preliminary searches of the literature were conducted in March and April 2019 to determine if any previous scoping or systematic reviews had been conducted or protocols submitted which aimed to summarise existing evidence on overdiagnosis and overtreatment in ADHD. Databases searched were the Cochrane Database of Systematic Reviews, PROSPERO, Medline, Scopus and the JBI (Joanna Briggs Institute) Database of Systematic Reviews and Implementation Reports. Two scoping reviews on overdiagnosis in healthcare were found, but neither review was focused on overdiagnosis and overtreatment of ADHD. One review covered the drivers of overdiagnosis and potential solutions,[22] while the other addressed overdiagnosis across different medical disciplines.[24 25] While a number of systematic reviews have been conducted on ADHD, these have been restricted to prevalence[26–28] or treatment options.[29 30] We also identified one systematic review on overprescribing or underprescribing of ADHD medication that is currently under way with a published protocol.[31] Literary reviews and commentaries on overdiagnosis in mental health, including ADHD, have been published.[13 21 32] However, we found no reviews that have been conducted or are in preparation that systematically gather and analyse all available evidence to allow a comprehensive assessment on the potential for overdiagnosis of ADHD.

### METHODS AND ANALYSIS

The proposed review will follow the Joanna Briggs Methodology for Scoping Reviews,[33] which is based on and extends the work of Arksey and O'Malley[34] as well as that of Levac and colleagues.[35] The scoping review will also adhere to the Preferred Reporting Items for Systematic Reviews and Meta-Analyses Extension for Scoping Reviews (PRISMA-ScR)[36] (online supplement I). This approach was chosen to document the review process as clearly and rigorously as possible. The timeframe for undertaking of the review is from 13 June 2019 (date that the final search strategy was run in all included databases) until 31 December (anticipated completion date of the review).

## Inclusion criteria

### Participants

The scoping review will include studies whose main focus is on children and adolescents under the age of 18 who have been either clinically diagnosed or identified by the parent, teacher or self-report as having behavioural symptoms of ADHD. Articles that have a clear emphasis on adult ADHD will be excluded unless they are longitudinal follow-up studies where participants were determined to have ADHD in childhood and were then followed through into adulthood. Moreover, studies with other health issues or disabilities as the primary focus will also be excluded. However, studies considering any other comorbidities as a secondary diagnosis will be eligible for inclusion if the focus of the study is clearly on ADHD (outcomes on any other disorders will not be reported or included in the analysis).

### Concept

The two core concepts to be examined by this scoping review are 'overdiagnosis of ADHD' and 'overtreatment of ADHD'. Debate over the definition of overdiagnosis is ongoing,[1 16 17 37 38] especially in non-cancer contexts. We define overdiagnosis as occurring where a person is correctly diagnosed (according to contemporary professional standards) with a condition but the net effect of the diagnosis for the individual concerned is unfavourable (ie, when consideration is given to the balance of potential harms and benefits).[1 39] The resulting overtreatment can then be defined as receiving treatment following an overdiagnosis (or among overdiagnosed individuals).[16 40] It is important to note that overtreatment can occur as a result of overdiagnosis but also without it, due to other drivers.[16] In this scoping review we will only consider pharmaceutical treatment options for ADHD in terms of overtreatment.

### Context

The study will be conducted in the context of a previously published and tested framework of five questions that identify characteristics consistent with the occurrence of overdiagnosis and overtreatment.[1] It is not limited to any geographical areas or settings. The five questions were developed by experts in the field of overdiagnosis as a guide to identifying areas in medicine where overdiagnosis and overtreatment may be occurring. Additionally, an overarching primer question will be included to avoid missing evidence. Hence, the questions by which the search will be guided and to which the evidence will be mapped are the following:

0. Is ADHD overdiagnosed in children and adolescents?
1. Is there potential for increased diagnosis?
2. Is diagnosis actually increased?
3. Are additional cases subclinical or low risk?
4. Are additional cases treated?
5. Might harms outweigh benefits?
   a. For treatment.
   b. For diagnosis.

### Types of evidence

For questions 0–4 any existing, peer-reviewed primary studies as well as systematic reviews will be considered. As our preliminary searches came up with very large amounts of evidence from primary studies that could be deemed suitable to answer question 5, we will limit the types of evidence included to answer this question as follows: the search for question 5 part A will be limited to systematic reviews (of randomised controlled trials or observational studies) and cohort studies investigating short-term and long-term outcomes from ADHD pharmaceutical treatment. The search for question 5 part B will be wider and include any primary studies investigating outcomes after an ADHD diagnosis.

### Search strategy

Initially, various basic, restricted searches in Medline and Scopus were performed on each of the five predetermined questions (1–5) to uncover some articles relevant to the topic. This initial search was followed by an analysis of key concepts in titles and abstracts as well as index/medical subject heading terms from key papers. Moreover, we reviewed published search strategies from reviews on similar topics to identify keywords related to our study aims.[24 25] A second full and complex search strategy with the identified keywords and index terms was then developed with the assistance of a research information specialist. This search will be conducted in Medline, Embase, PsycINFO and the Cochrane Library to locate articles relevant to all five aspects of the framework and the additional primer question (0). For practical reasons this search will be restricted to English-language articles only. Databases will be searched from 1979 onwards. Publications from before 1979 will be excluded as their findings would reflect a historical definition of ADHD (hyperkinetic reaction of childhood) in line with DSM-II.[41] These are unlikely to be relevant to our study question. The complete search strategy for Medline can be found in online supplement II. Finally, the search will be supplemented by forward and backward citation searches of all included papers.

### Study selection

After an initial pilot phase to ensure appropriate training for high-level decision making and to test our inclusion/exclusion criteria, all titles and abstracts identified by the database and hand searches will be screened and reviewed for relevance by two researchers independently. Abstrackr (http://abstrackr.cebm.brown.edu), a text mining tool, will be used to help with this initial screening.[42 43] Full-text reviews of all potentially suitable papers will be independently conducted by two researchers, according to the predefined inclusion/exclusion criteria. All studies excluded at the full-text screening stage will be reported with reasons for exclusion provided. At both stages of the screening process (abstract and full-text screening), any discrepancies will be resolved through discussion with the team.

## Data extraction

Data from the final articles to be included in the review will be charted independently into a standardised and piloted template by two researchers. Any uncertainties will be again discussed and resolved by the entire study team. Data will be extracted on the source, eligibility, methods, population characteristics, intervention/exposure, outcomes, results and other areas of interest. A template for data extraction is attached (online supplement III) and may be further refined and updated during the review stage. As part of the data extraction process, all included studies will undergo a basic critical appraisal using the Joanna Briggs Institute Critical Appraisal Tools for the relevant study type (https://joannabriggs.org/critical_appraisal_tools).

## Presentation of results

All information regarding the selection of sources will be presented in a flow diagram according to PRISMA-ScR.[36]

Results from all included studies will be mapped to our predetermined five-question framework in the form of a table as well as in descriptive, narrative form. This results table will include summary information from the conducted critical appraisals. Estimates from quantitative studies will be included in a summary table but will not be meta-analysed as we expect results to be too heterogeneous to allow for meaningful synthesis. Further, evidence will be categorised by type of article and study type in order to highlight where additional research may be needed to fill current evidence gaps.

## Patient and public involvement

Neither the protocol nor the proposed scoping review will involve patients or members of the general public.

## ETHICS AND DISSEMINATION

Due to the proposed study being a scoping review, there are no ethical or safety considerations to be made. It is planned to disseminate the results from the scoping review through publication in a peer-reviewed journal and through conference presentations.

**Twitter** Luise Kazda @@LuiseKazda, Katy Bell @KatyJLBell and Rae Thomas @rthomasEBP

**Acknowledgements** The authors acknowledge the contributions towards the search strategy development and other useful recommendations with regard to literature searching made by Justin Clark, Senior Research Information Specialist at the Institute for Evidence-Based Healthcare, Bond University.

**Contributors** LK, KB, RT and AB contributed to the conception and design of the protocol. LK, KB, RT, KM and AB contributed to the establishment of searches. LK drafted the protocol, KB, RT, KM and AB made contributions to the drafting and revising of the article. All authors approved the final version of the protocol for publication and its accuracy and integrity.

**Funding** This work is supported by Wiser Healthcare, which is funded by the National Health and Medical Research Council (NHMRC) Program Grant 1113532 and CRE Grant 1104136. The funding source has no role in study design, data collection, data analysis, data interpretation or writing of the report.

**Competing interests** None declared.

**Patient consent for publication** Not required.

**Provenance and peer review** Not commissioned; externally peer reviewed.

**ORCID iDs**
Luise Kazda http://orcid.org/0000-0003-4105-0402
Katy Bell http://orcid.org/0000-0002-0137-3218
Rae Thomas http://orcid.org/0000-0002-2165-5917

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
