## [Reviewer comments · BMJ Open]

ARTICLE DETAILS

TITLE (PROVISIONAL)	Evidence of potential overdiagnosis and overtreatment of Attention Deficit Hyperactivity Disorder (ADHD) in children and adolescents: protocol for a scoping review
AUTHORS	Kazda, Luise; Bell, Katy; Thomas, Rae; McGeechan, Kevin; Barratt, Alexandra

VERSION 1 – REVIEW

REVIEWER	Samuele Cortese University of southampton
REVIEW RETURNED	24-Jul-2019

GENERAL COMMENTS	The paper addresses a very relevant questions in the field of ADHD. However, I have a number of comments/concerns: Title: I am not clear why the title mentions only “overdiagnosis and overtreatment” when the question is as to whether ADHD is under or over diagnosed/treated Strengths and limitation section: not clear what “in a new condition..” refers to Background Please explain why cancer, and not other medical conditions, is mentioned; this comes a little out of the blue I would suggest to rephrase “Overdiagnosis of ADHD appears to be a result of three main driver” as it is not clear if overdiagnosis occurs. I would rephrase as “a possible overdiagnosis of ADHD might be the result of three main driver DSM-V should be DSM-5 “Overdiagnosis and overtreatment also results in financial and opportunity costs to the health system, and to society at large”: the harms of underdiagnosing and undertreating should also be highlighted Methods I am not clear how specifically and practically speaking, the following key questions will be addressed based on data from the retrieved papers: Is ADHD overdiagnosed in children and adolescents? Is there potential for increased diagnosis? Is diagnosis actually increased? Are additional cases subclinical or low risk? Are additional cases treated? Might harms outweigh benefits? (a) for treatment, (b) for diagnosis
---

	If, for instance, there were two meta-analysis, once concluding that ADHD is under treated and another showing that it is overtreated, how will the authors deal with this situation? “Quantitative estimates will be included in our synthesis of results where applicable and meaningful” please elaborate further. How will quantitative estimates form a number of different SR/MSA be included and when will they be considered meaningful?
--	--

REVIEWER	Michael J. Manos, PhD Cleveland Clinic Cleveland, OH U.S.A
REVIEW RETURNED	05-Aug-2019

GENERAL COMMENTS	A scoping review is quite a complex exercise. To really address the question of overdagnosing and overtreating ADHD, one must clearly cite the inadequacy of methods used to diagnose ADHD and then cite the incidence of children not symptomatic of the disorder that have been nevertheless treated. My unfamiliarity with the complexities of a scoping review may have me err in these comments, however. The objectives of the proposed study appear to be attainable, nevertheless, and will require an immense amount of work as contrasted with the methodology of a meta-analysis. The results may not carry the weight of or be as definitive as a well-done meta-analysis so the authors must be sure to cite a large enough N and clearly state the limitations of the methodology in the write-up of results. Given the importance of the question, this would be a useful, certainly complex study, to undertake. Thus, in the results write up, the authors would do well to cite the limitations of the methodology.
--

REVIEWER	Ramesh Lamsal University of Toronto, Canada
REVIEW RETURNED	31-Aug-2019

GENERAL COMMENTS	I read your manuscript with interest. Following recommendations will improve the paper. Describe sources or process of "search terms." (from existing reviews on similar topics, consultation with experts ADHD often co-occurs with other symptoms and disorders, including ASD, anxiety, learning disabilities, etc. Although researchers have stated that they are "considering other mental health co-morbidities as a secondary diagnosis will be eligible for inclusion." I would recommend listing all disorders that they are planning to include. Are authors also planning to report or analyze potential over-diagnosis of other co-morbidities? For instance, a child may have been diagnosed with ASD and ADHD, but only overdiagnosed with ADHD. I would strongly recommend adding a basic critical appraisal tool (CAT) or quality assessment tool to examine the quality of the included studies, how you will critically appraise the quality of the evidence from each of your studies?
---

VERSION 1 – AUTHOR RESPONSE

REVIEWER 1 COMMENTS & RESPONSES

1. The paper addresses a very relevant questions in the field of ADHD.

Response: We thank the reviewer for this supportive general comment and for his helpful feedback.

1.1 Title: I am not clear why the title mentions only “overdiagnosis and overtreatment” when the question is as to whether ADHD is under or over diagnosed/treated

Response: As stated in the rationale, our aim with this study is “...to systematically determine if the existing literature indicates a potential for overdiagnosis and overtreatment in ADHD.” Thus, we feel it is appropriate to exclude underdiagnosis and undertreatment from the title. Whilst we are aware that there is also considerable debate in the field about underdiagnosis of ADHD, we believe that this is a separate issue independent of whether or not there is overdiagnosis/overtreatment and that this is beyond the scope of our review. We have amended the title of the manuscript to better reflect our study aim. We have also added a statement in the Background Section on the issue of underdiagnosis and we will make sure to include a comment in the Discussion Section of the final study as a recommendation for further research. Moreover, we have made it clear in the Strengths and Limitations Section that the study does not consider underdiagnosis/undertreatment (also see comment 0.2).

Change in paper: Added the following two sentences into the Background Section referring to underdiagnosis (p 4/5): *“It has also been pointed out that the resulting potential overuse of healthcare resources contributes to the simultaneous underuse of said resources²²; e.g. by depriving underdiagnosed and undertreated groups of children who would largely benefit from ADHD treatment and timely access to diagnostic and treatment services²³. Whilst also addressing the twin issues of underdiagnosis and undertreatment is beyond the scope of this proposed review, we see our work as a starting point for a broader discussion around the principles of “right care” where resources need to be re-allocated to where they are most needed and effective²².”*

Added a limitation of the study in the Strengths and Limitations Section (p 3): *“Underdiagnosis and undertreatment may also occur, but are outside the scope of this review.”*

1.2 Strengths and limitation section: not clear what “in a new condition” refers to

Response: Thank you, this is indeed misleading, we have changed the entire Strengths and Limitations Section (see comment 0.2) and have taken out this statement completely.

Change in paper: Removed and replaced the sentence in the “Strengths and Limitations” sections (p 3), also see comments 0.2 and 1.1.

1.3 Background: Please explain why cancer, and not other medical conditions, is mentioned; this comes a little out of the blue

Response: Methodological research on overdiagnosis has its beginnings (and is more established) in cancer (asymptomatic condition, e.g. thyroid, prostate, breast cancer) where it is easier to define and describe than in non-cancer conditions (often symptomatic conditions) (Carter BMJ 2015). Our reference to these conditions meant to contrast them to a neurodevelopmental condition like ADHD, where, so far, there has been much speculation on overdiagnosis but minimal quantitative evaluation to examine the extent of this. We have made amendments to this section to make the connection clearer.

Change in paper: Changed “*Whilst the evidence for overdiagnosis in many cancers is increasingly recognised...*” to “*Whilst the evidence for overdiagnosis in many other conditions (especially in screen detected cancers) is increasingly recognised...*” in the Background Section (p 4). We have also added another reference to this sentence for explanation (Bell et al).

1.4 Background: I would suggest to rephrase “Overdiagnosis of ADHD appears to be a result of three main driver” as it is not clear if overdiagnosis occurs. I would rephrase as “a possible overdiagnosis of ADHD might be the result of three main drivers

Response: We agree with this suggestion and have amended the text accordingly.

Change in paper: Changed “*Overdiagnosis of ADHD appears to be a result of three main drivers.*”, to “*The potential overdiagnosis of ADHD could be caused by three main drivers.*” In the Background Section of the manuscript (p 4).

1.5 Background: DSM-V should be DSM-5

Response: This error has been corrected.

Change in paper: Changed “*DSM-V*” to “*DSM-5*” in the Background Section (p 4).

1.6 Background: “Overdiagnosis and overtreatment also results in financial and opportunity costs to the health system, and to society at large”: the harms of underdiagnosing and undertreating should also be highlighted

Response: We agree that underdiagnosis and undertreating may also be potentially harmful, and although not the focus of our review, we now highlight the fact that financial and opportunity costs of overdiagnosis and overtreatment may actually cause underdiagnosis and undertreatment of children who would benefit from treatment.

Change in paper: See change in 1.1 (p 4/5).

1.7 Methods: I am not clear how specifically and practically speaking, the following key questions will be addressed based on data from the retrieved papers: (...) If, for instance, there were two meta-analysis, once concluding that ADHD is under treated and another showing that it is overtreated, how will the authors deal with this situation?

Response: It is important to note that overdiagnosis/overtreatment and underdiagnosis/undertreatment are not mutually exclusive and are likely to co-exist. As above (1.1 and 1.6), this review will not include primary or review studies with a focus on underdiagnosis

of ADHD as its central question is on the synthesis of evidence for overdiagnosis. Thus, from the evidence we find, we can only conclude that there is or is not enough evidence to indicate overdiagnosis in ADHD. Moreover, as stated in our Rationale Section, we are not aware of any existing reviews summarising evidence on the overdiagnosis/-treatment of ADHD; hence the need for this review.

Change in paper: See change in 1.1.

1.8 Methods: “Quantitative estimates will be included in our synthesis of results where applicable and meaningful” please elaborate further. How will quantitative estimates form a number of different SR/MSA be included and when will they be considered meaningful?

Response: As this is a scoping review we do not plan on a quantitative synthesis/meta-analysis of results. We anticipate identifying a very heterogenous range of study types that will not allow us to combine their results in any sort of meta-analysis. Hence, we will present our main results in a narrative summary as described. We also anticipate including the quantitative estimates from our sources in a table. These will not be combined or summarised and will only be applicable where original studies are of quantitative nature. We have made some amendments in the text to make this clearer.

Change in paper: Made changes in the Methods - Presentation of Results Section accordingly (p 8). Changed “*Quantitative estimates will be included in our synthesis of results where applicable and meaningful.*”, to “*Estimates from quantitative studies will be included in a summary table but will not be meta-analysed as we expect results to be too heterogenous to allow for meaningful synthesis.*”

REVIEWER 2 COMMENTS & RESPONSES

2.1 A scoping review is quite a complex exercise. To really address the question of overdiagnosing and overtreating ADHD, one must clearly cite the inadequacy of methods used to diagnose ADHD and then cite the incidence of children not symptomatic of the disorder that have been nevertheless treated. My unfamiliarity with the complexities of a scoping review may have me err in these comments, however. The objectives of the proposed study appear to be attainable, nevertheless, and will require an immense amount of work as contrasted with the methodology of a meta-analysis. The results may not carry the weight of or be as definitive as a well-done meta-analysis so the authors must be sure to cite a large enough N and clearly state the limitations of the methodology in the write-up of results. Given the importance of the question, this would be a useful, certainly complex study, to undertake. Thus, in the results write up, the authors would do well to cite the limitations of the methodology.

Response: We thank the reviewer for his comments on our protocol. As noted, we are also very aware of the large undertaking of such a broad scoping review (as opposed to a more focussed systematic review) but felt this was the only feasible way to summarise all the relevant evidence

on the topic of overdiagnosis in ADHD due to the limitations in primary studies available. We certainly anticipate being able to reference a very large number of relevant studies, which will allow us to give the review as much weight as is possible for this type of study. We also agree with his recommendation to clearly state the limitations of this study type in the Discussion Section of our final results.

Change in paper: None

REVIEWER 3 COMMENTS & RESPONSES

3. I read your manuscript with interest. Following recommendations will improve the paper.

Response: We thank the reviewer for this supportive general comment and for his helpful feedback.

3.1 Describe sources or process of "search terms." (from existing reviews on similar topics, consultation with experts)

Response: We have made amendments to the Methods - Search Strategy Section of the paper to include some more specific details on the sources of search terms used in the review.

Change in paper: Added more details to the Methods - Search Strategy Section in this sentence (p 7): *"Moreover, we reviewed published search strategies from reviews on similar topics to identify key words related to our study aims^{24, 25}."*

3.2 ADHD often co-occurs with other symptoms and disorders, including ASD, anxiety, learning disabilities, etc. Although researchers have stated that they are "considering other mental health co-morbidities as a secondary diagnosis will be eligible for inclusion." I would recommend listing all disorders that they are planning to include. Are authors also planning to report or analyze potential over-diagnosis of other co-morbidities? For instance, a child may have been diagnosed with ASD and ADHD, but only overdiagnosed with ADHD.

Response: We are aware of the high co-morbidity of ADHD with numerous other disorders. However, the focus of this review will be on the overdiagnosis of ADHD only. Whilst we will not automatically exclude studies that list other disorders alongside ADHD, we will only be reporting on the relevant ADHD-related outcomes. Concentrating this review on one disorder allows us to gain a better in-depth understanding of the mechanisms involved in overdiagnosing/overtreating children with this specific condition. This knowledge may then be translated to other mental health or neurodevelopmental conditions in further research. We have amended the Methods - Inclusion Criteria Section of the text to better reflect this intention.

Change in paper: Changed a sentence in the Methods - Inclusion Criteria Section (p 6) from "... studies considering other mental health comorbidities as a secondary diagnosis will be eligible for inclusion.", to "... studies considering any other comorbidities as a secondary diagnosis will be eligible for inclusion if the focus of the study is clearly on ADHD (outcomes on any other disorders will not be reported on or included in the analysis)."

3.3 I would strongly recommend adding a basic critical appraisal tool (CAT) or quality assessment tool to examine the quality of the included studies, how you will critically appraise the quality of the evidence from each of your studies?

Response: As our proposed review is intended as a scoping review and we expect to find a large number of heterogenous study types to be included in the final review, we have decided not to include a formal risk of bias analysis. However, we agree with the reviewer's recommendation that a basic critical appraisal tool would be of benefit and strengthen the study results. Since we are basing our review on the JBI methodology for scoping reviews, we will be using the Joanna Briggs Institute Critical Appraisal Tools (https://joannabriggs.org/critical_appraisal_tools) for a basic appraisal of all included studies.

Change in paper: Added a sentence with details in the Methods - Data Extraction Section (p 8): *"As part of the data extraction process all included studies will undergo a basic critical appraisal using the Joanna Briggs Institute Critical Appraisal Tools for the relevant study type (https://joannabriggs.org/critical_appraisal_tools)."* and one in the Presentation of Results Section (p 8): *"This results table will include summary information from the conducted critical appraisals."*

VERSION 2 – REVIEW

REVIEWER	Samuele Cortese University of Southampton
REVIEW RETURNED	08-Oct-2019
GENERAL COMMENTS	Thank you for addressing my concerns
REVIEWER	Ramesh Lamsal University of Toronto, Canada
REVIEW RETURNED	05-Oct-2019
GENERAL COMMENTS	The manuscript has been largely improved. No further comments. Thank you for making changes.